# Extreme precipitation and flooding in Berlin under climate change and effects of selected grey and blue-green measures

Franziska Tügel<sup>1,3,4</sup>, Katrin M. Nissen<sup>2</sup>, Lennart Steffen<sup>1</sup>, Yangwei Zhang<sup>1</sup>, Uwe Ulbrich<sup>2</sup>, and Reinhard Hinkelmann<sup>1</sup>

**Correspondence:** Franziska Tügel (franziska.tuegel@utwente.nl)

20

**Abstract.** This paper aims to quantify potential changes in extreme precipitation under climate change scenarios in the city of Berlin, Germany, and their resulting impacts on urban flooding in a selected flood-prone area of the city. Furthermore, it investigates the effectiveness of the existing drainage system, infiltration from unsealed surfaces, and retention roofs during extreme rainfall events under both current and future climate conditions. Finally, uncertainties in infiltration are addressed by varying soil hydraulic conductivity and the degree of surface sealing.

The effect of climate change on the statistical distribution of extreme precipitation in Berlin is assessed by analyzing a single-model set of climate scenario simulations at convection permitting resolution (COSMO-CLM). Three 30-year periods are simulated: The historical period under observed greenhouse gas concentrations from 1971 to 2000 and two RCP8.5 scenario periods from 2031 to 2060 and from 2071 to 2100. For the historical period, the estimated 1-hour rainfall sum for a 100-year return level (referred to as 'Historical 100a') agrees well with the statistical values from station observations. For the period 2031-2060 under RCP8.5 conditions the respective rainfall sum of the 1-hour 100-year event (referred to as 'Future 100a') increases by 46% and the strongest hourly intensity in all three simulated 30-year periods (referred to as 'Strongest') is increased by 123% compared to the Historical 100a event.

The impacts of these increases in extreme precipitation on flooding characteristics in a Central-Berlin region around the Gleimtunnel, which is known for frequent pluvial flooding, are studied by conducting simulations with the 2D surface flow model hms<sup>++</sup> coupled to a 1D drainage model. The Future 100a event results in a 51% increase in the simulated maximum water depth, a 43% increase in maximum surface runoff at the local flooding hotspot Gleimtunnel, and a 33% increase in the volume of combined sewer overflow at one selected outfall. For the Strongest event, the respective increases are 137% (maximum water depth), 296% (maximum surface runoff), and 74% (combined sewer overflow).

The effects of the existing drainage system and infiltration under different rainfall scenarios are highlighted by comparing simulation results with and without their consideration. Neglecting the drainage system results in a 170% increase in the maximum water depth at the Gleimtunnel for the Historical 100a event and a 110% increase for the Strongest event, compared to

<sup>&</sup>lt;sup>1</sup>Chair of Water Resources Management and Modeling of Hydrosystems, Technische Universität Berlin, Berlin, Germany <sup>2</sup>Institute for Meteorology, Freie Universität Berlin, Berlin, Germany

<sup>&</sup>lt;sup>3</sup>Department of Water Resources, Faculty of Geo-Information Science and Earth Observation (ITC), University of Twente, Enschede, The Netherlands

<sup>&</sup>lt;sup>4</sup>Multidisciplinary Water Management, Civil Engineering and Management, Faculty of Engineering Technology, University of Twente, Enschede, The Netherlands

the reference simulations. While the drainage system strongly reduces flooding, especially at hotspots, it cannot fully prevent severe flooding, and its effectiveness decreases with higher rainfall intensity. Studying infiltration reflects potential impacts of surface sealing or, conversely, desealing as a climate adaptation strategy. Neglecting infiltration increases the maximum water depth at the Gleimtunnel by 33% for the Historical 100a event and 18% for the Strongest event compared to the reference simulations. The volume of combined sewer overflow increased by 19-30% if infiltration was neglected. Infiltration significantly reduces flooding, though its effectiveness decreases with higher rainfall intensity.

As a potential adaptation strategy, the impact of replacing all roofs with retention roofs is examined. For this best-case adaptation scenario, the maximum water depth at the local hotspot is reduced by 22-24%, and the volume of combined sewer overflow by 15-20% in the different scenarios. Since full retention on all roof surfaces is considered for all rainfall scenarios, the effects are almost the same. Remarkably, the retention roofs significantly reduced the maximum surface runoff in the Gleimtunnel during the Strongest event to below the stability threshold for pedestrians, which was clearly exceeded in the simulation without retention roofs.

The results of this study highlight the potential local impacts of ongoing global warming in terms of heavy rainfall and urban flooding in the city of Berlin and emphasize the need to combine grey infrastructure, retention roofs, surface desealing, and other blue-green measures.

## 1 Introduction

35

While there is a European-wide obligation to produce risk maps for river floods (EUR-Lex, 2007), this is not the case for pluvial flooding. Nevertheless, such maps are being produced for more and more cities, including Berlin (Umweltatlas Berlin). Information maps for hotspots of heavy precipitation impact already exist for the whole city. These show ground depressions and fire brigade operations required in the past because of flooding. More detailed risk maps for pluvial flooding are currently only publicly available for 3 smaller areas within the city. These maps show the spatial extent of flooding, the flood depths and the flow velocities for three different heavy rainfall scenarios and are generated with coupled 2D-1D hydrodynamic models. The Federal Agency for Cartography and Geodesy (BKG) is currently working with the federal states to develop Germany-wide risk maps for pluvial flooding. The maps for Berlin have become available in 2025 (BKG, 2025). The first subregion for the Germany-wide maps for pluvial flooding, North Rhine-Westphalia, has been available as an interactive web map since October 2021. The maps are based on 2D hydrodynamic simulations that simplify the system by not considering the underground drainage network or infiltration from permeable surfaces (BKG, 2021). Different precipitation scenarios categorized as rare, exceptional and extreme are used as input. The exceptional event is defined as a 100-year precipitation event (T = 100a) with a duration of one hour and an Euler type 2 (Euler-2) temporal evolution (e.g. Wartalska et al., 2020; DWA, 2006). The Euler-2 evolution is characterized by gradually increasing intensities, reaching the 5-minute maximum after 20 minutes, and a subsequent strong decrease down to a small residual amount (as illustrated in section 4.3, Fig.5). The time series is constructed using hourly and sub-hourly, statistical precipitation intensities from the coordinated heavy precipitation regionalization and evaluation (KOSTRA, 2020) of the German Weather Service (DWD; see section 2).

The problem with these statistical values is that they are based on historical observations and will be outdated soon according to climate simulations. Researchers agree in their assessment that over Central Europe extreme precipitation events will occur more frequently and become more intense in a warmer climate (IPCC, 2022). To investigate possible future developments for a specific region, climate models that are driven by plausible greenhouse gas (GHG) scenarios can be analyzed. Ideally, for investigations focusing on extreme precipitation events, regional models with convection-permitting resolution are used for this task (e.g. Lucas-Picher et al., 2021). To determine extreme value statistics, long simulations are needed. These are computationally expensive and usually cover only small areas. At the time of writing only one suitable set of simulations at convection permitting resolution covering Berlin was available. We use these simulations to analyze possible changes in the statistical distribution of extreme precipitation in Berlin. The simulation set includes three 30-year periods under historical and Representative Concentration Pathway (RCP) 8.5 GHG conditions.

In the next step, the impacts of changes in extreme precipitation on the characteristics of pluvial flooding in a selected urban area are investigated using hydrodynamic model simulations. These simulations are conducted with the 2D shallow water model hms<sup>++</sup>, which is bi-directionally coupled to the 1D Storm Water Management Model (SWMM). For example, Li et al. (2024) conducted similar investigations for an urban area in Haining, Zhejiang Province, China, using the SWMM model with the hydrological subcatchment approach, rather than coupling it with a 2D surface flow model. Neumann et al. (2024) carried out a detailed analysis of the effectiveness of different types of decentral nature-based solutions for flood reduction in another area of Berlin. While they also used a 2D-1D coupled modeling approach, roofs and measures were not represented as 2D surfaces, but through hydrological elements within SWMM. In our study, the focus is primarily on the effects of increased heavy precipitation on spatially distributed flood depths and surface runoff, temporal developments at a selected flooding hotspot, and combined sewer overflow. Furthermore, the effects of infiltration, the drainage system, and retention roofs as one example for blue-green infrastructure are studied. The following paragraphs briefly describe the basics of these processes and the motivation for studying their effects on urban flooding.

Infiltration is an important process in the water cycle. Rainfall intensities exceeding the infiltration capacity are generally considered as the reason for the formation of so-called Hortonian overland flow, which is often the type of overland flow during flash floods instead of saturated overland flow. In urban flood simulations, infiltration is sometimes neglected (e.g. Tügel et al., 2023), referring to the high degrees of surface sealing and choosing a conservative approach to "be on the safe side". But also in urban areas, understanding the effects of infiltration is crucial - for example, to estimate how further surface sealing could exacerbate flooding and to evaluate the potential flood reduction benefits of surface desealing.

The drainage system can be considered the most important grey infrastructure for mitigating surface flooding from rainfall. However, its dimensions are typically designed for return periods of only two to five years. In simulations of extreme rainfalls, the drainage system is sometimes neglected (Tügel et al., 2023) or considered with simplified approaches of reduced rainfall on sealed surfaces (Apel et al., 2024). In order to consider both the surface runoff and the flow in the drainage system, a bidirectionally coupled 2D-1D model is needed. To quantify the effectiveness of the drainage system on the reduction of surface flooding even for extreme precipitation events, this study compares simulation results with and without the consideration of the drainage system.

A coupled approach is also preferred as it allows the simulation of overflow in cases where the drainage system is overstrained, which can exacerbate surface flooding at low-lying locations. In the case of a combined sewer system this leads to
severe pollution and hygienic problems in addition to flooding. In a combined sewer system rainwater and waste water share
the same pipes, which is the case in the central districts of Berlin including the considered study area. While overflows from
the drainage to the urban surface area usually only occur during very heavy events, combined sewer overflows of untreated
mixed rain and sewage water into water bodies occur many times a year in Berlin causing severe environmental pollution. In
this study we will also quantify the effect of different extreme precipitation scenarios as well as infiltration and retention roofs
on the combined sewer overflow.

In recent years to decades there is a paradigm shift from the aim of "transporting the rainwater as fast as possible out of the city" to "keeping as much rainwater as possible in the city". This should be achieved for example with blue-green infrastructure. The goal is to restore a more natural water balance in cities by decreasing surface runoff and flooding, increasing evapotranspiration and groundwater recharge and/or saving drinking water by using captured rainwater for non-potable water uses (Zevenbergen et al., 2018). Climate change is increasing the pressure on water resources, which makes this shift in the urban water management inevitable. Decoupling surfaces and holding back more rainwater inside the urban area should also relieve the drainage system itself and reduce the combined sewer overflows in case of heavy rain events. Blue-green infrastructure offers several benefits and can also contribute to reducing urban flooding.

## 2 Data

105

# 2.1 Meteorological observations

KOSTRA-DWD (KOSTRA, 2020) is the official source for statistical extreme precipitation intensities in the time period 19512020. KOSTRA-DWD is compiled by the German Weather Service (DWD) on the basis of gridded precipitation observations. It
lists the precipitation intensities of heavy rainfall events in terms of their duration and statistical return period. For the analysis,
a duration dependent general extreme value distribution (d-GEV) is fitted to the annual maxima of the gridded observations
(Junghänel et al., 2022; Koutsoyiannis et al., 1998). A key area of application is the dimensioning of stormwater management
structures such as sewer networks, pumping stations, and retention basins, and they are also used to create flood maps for
pluvial flooding. For comparisons with measurements at a local station, data from the weather station Berlin Dahlem is used.

## 2.2 Climate simulations

For this study precipitation for present day and climate change conditions is taken from simulations of the regional climate model COSMO-CLM at convection permitting resolution (CCLM-CPS). The simulations have been described in detail by Rybka et al. (2023). They are conducted at a horizontal resolution of 0.0275° (approx. 3 km). Deep convection is explicitly solved, while shallow convection is parameterized. Precipitation output is available at 1-hour aggregation period. The simulations cover Germany and surrounding regions in order to capture the the river basins that discharge towards Germany. The

forcing data stems from the global climate model MIROC5 (Model for Interdisciplinary Research on Climate, Watanabe et al., 2011). CCLM-CPS simulations were conducted for 3 30-year periods: The historical period 1971 to 2000 and the RCP8.5 scenario periods 2031 to 2060 (near future) and 2071 to 2100 (far future) (Haller et al., 2022a, b). The ability of CCLM-CPS to simulate extreme precipitation intensities has been analyzed by Rybka et al. (2023). It was found that CCLM-CPS for the historical period shows the best agreement with the KOSTRA estimate compared to several other regional model ensembles.

## 2.3 Ground properties

125

130

For the hydrodynamic simulations, the digital elevation model provided by ATKIS® DGM Berlin with 1 m spatial resolution was used, incorporating buildings based on ALKIS Berlin/Gebäude. Land use types are based on the dataset ALKIS Berlin/Tatsächliche Nutzung. Fields of the saturated hydraulic conductivity of the first 1 m of soil were taken from the dataset Umweltatlas Berlin (2019) and Umweltatlas Berlin (2024). The sealing degrees of block and street areas are taken Umweltatlas Berlin (2022). These datasets are licensed under dl-de/by-2-0 (www.govdata.de/dl-de/by-2-0). The model of the subsurface drainage system was provided by Berliner Wasserbetriebe (BWB). Table 1 lists all used spatial datasets that are all publicly available.

**Table 1.** Overview of used spatial datasets for the hydrodynamic model setup (licensed under dl-de/by-2-0 www.govdata.de/dl-de/by-2-0; links are provided in the reference list)

| Data                         | Source                            | <b>Spatial Resolution</b> | Field (for vectorlayers) |
|------------------------------|-----------------------------------|---------------------------|--------------------------|
| Land use                     | ALKIS Berlin/Tatsächliche Nutzung | Block subareas            | BEZEICH                  |
| Sat. hydr. conduct. 2015     | Umweltatlas Berlin (2019)         | Block areas               | kf                       |
| Sat. hydr. conduct. 2020     | Umweltatlas Berlin (2024)         | Block areas               | kf                       |
| Surface sealing block areas  | Umweltatlas Berlin (2022)         | Block areas               | provgneu_2021            |
| Surface sealing street areas | Umweltatlas Berlin (2022)         | Street polygons           | provgneu_0               |
| Digital elevation model      | ATKIS® DGM Berlin                 | Raster 1 m                | -                        |
| Buildings                    | ALKIS Berlin/Gebäude              | Building polygons         | -                        |

#### 135 3 Methods

### 3.1 Return levels

Return levels are estimated using the method described by Fauer et al. (2021), which is an extension of the method used for KOSTRA-DWD. For each grid box located within the boundaries of Berlin, the annual maximum intensities (z) for different accumulation periods (durations d) are extracted from the climate simulations. The values for d=1 h are shown exemplary in Fig. 1. The probability distribution is modelled by fitting a duration dependent general extreme value distribution (d-GEV)

**Figure 1.** Annual maximum of hourly precipitation in CCLM-CPS simulation for all grid boxes in Berlin. Horizontal lines and corresponding labels denote the median over the simulation period. The Strongest event simulated is marked by an asterisk.

to these maxima (Eq. 1). Five parameters need to be determined: The rescaled location parameter  $\tilde{\mu}$ , the scale offset  $\sigma_0$ , the duration offset  $\Theta$ , the duration exponent  $\eta$ , and the shape parameter  $\xi$ .

$$G(z) = exp\left\{ -\left[1 + \xi \left(\frac{z}{\sigma_0(d+\Theta)^{-\eta}}\right)\right]^{-1/\xi}\right\}. \tag{1}$$

For durations > 1 hour the resulting Intensity-Duration-Frequency (IDF) curves follow a power law, and the duration offset  $\Theta$ , which describes the curvature, is negligible.

Our hypothesis is that the underlying distribution of the extreme precipitation at the horizontal resolution resolved by station observations and by the climate simulations does not exhibit spatial variations within Berlin and its immediate vicinity. We therefore assume that the spatial variations between neighboring grid boxes are a random effect caused by the limited number of observations (shortness of the time series). This is supported by the findings of Rybka et al. (2023) showing no indication for systematic patterns in observed extreme precipitation for both daily and hourly events in our study area. By pooling the data over all grid boxes located within the boundaries of Berlin we are able to include more data in the parameter estimation of the d-GEV distribution. This reduces the uncertainty associated with the parameter estimation and increases the robustness of the inferred climate change signal. The sampling uncertainty is determined using a bootstrap approach that randomly draws years with replacement.

The fitting of the d-GEV distribution is conducted using the R-package IDF (Ulrich et al., 2020), which determines the parameters by minimizing the negative log-likelihood.

## 3.2 Hydrodynamic modeling

160

To investigate urban flooding generated by heavy rainfalls from climate simulations as well as effects of different mitigation measures, a 2D-1D hydrodynamic model is set up for one selected sewer drainage catchment within the city of Berlin. The open-source 2D shallow water model hms<sup>++</sup> is used for the simulation of surface flow. It numerically solves the depth-averaged shallow water equations using a robust MUSCL scheme based on the finite volume method. Furthermore, the performance of the model has been optimized to achieve shorter computational times, and methods for further performance optimizations are currently under development. Details can be found in Simons (2020) and Steffen and Hinkelmann (2023). hms<sup>++</sup> is bidirectionally coupled to the 1D drainage model of the open-source Storm Water Management Model (SWMM) (Rossman and Huber, 2015). The surface model spans an area of approximately 13 km<sup>2</sup>. It consists of 543 900 rectangular cells of 5 m cell length. A spatial resolution of 5 m was selected as a suitable compromise between computational effort and accuracy of the numerical simulations. A finer resolution of 2 m was tested in a sensitivity study (see section 4.3).

The drainage model was provided by the Berlin Water Company (BWB) consisting of overall 24 km pipe lengths including 1964 nodes and 2344 links. The system includes pumps, orifices, and weirs. The maximum depth of the conduits varies between 0.12 and 2 m. The Manning's friction coefficient of all conduits is 0.013 s/m<sup>1/3</sup>, the entry loss coefficient 0.15 and the exit loss coefficient 0.015.

For the surface flow model, the digital elevation model based on ATKIS® DGM Berlin 1 m was modified to cut free the Gleimtunnel and incoporate buildings by increasing the elevation by 10 m in polygons based on ALKIS Berlin/Gebäude. Spatially distributed friction and infiltration values were generated based on soil data, land use, and sealing degrees from publicly available datasets. Manning's friction coefficients for the different land use types according to ALKIS Berlin/Tatsächliche Nutzung were selected from recommendations given in Klimaatlas NRW (2024). Infiltration was considered with spatially distributed and temporally constant infiltration capacity calculated based on the saturated hydraulic conductivity from the first 1 m of soil provided in the dataset Umweltatlas Berlin (2019) multiplied with the unsealing degree calculated from sealing degrees of the unbuilt surfaces on a block area level provided in the dataset Umweltatlas Berlin (2022). The average sealing degree of the unbuilt surfaces is about 26%. Buildings and for most simulations also street areas were considered as completely sealed with an infiltration rate of zero, which is the case for 51% of the considered model domain consisting of 35% roof and 16% road surfaces. In the model, no roof drainage directly into the sewer system is represented. Since buildings are characterized by their elevated position, rain falling on the roofs flows off the buildings, either into backyards or onto street surfaces, from where it eventually enters the sewer system via the street inlets. In section 4.3.2, retention roofs are represented in a simplified manner by setting the rainfall input to zero for all roof surfaces. This assumes full retention of all rainwater falling on the roofs. The model is strongly based on physical relationships and available data for estimating model parameters. Model validation of the surface flow model was carried out by comparing results for other parts of the city with official flood maps from the municipality (Geoportal Berlin/Starkregen) as well as with simulations from other projects on urban flooding in Berlin (AMAREX; InnoMAUS). For the coupled model, plausibility checks have been carried out for selected heavy rain events comparing simulated time series of the water depth in a retention tank with observed data. The largest uncertainties might be associated to infiltration processes. In section 4.3.4, the constant infiltration capacity and degree of surface sealing are varied to assess their effect on the results. Fig. 2 depicts the elevations including buildings, infiltration rates, and Manning's friction coefficients for the study area as well as its location within Berlin. The total simulation time was two hours, with precipitation occurring during the first hour, as only events with a 1-hour duration were selected for the hydrodynamic simulations, focusing on short, intense events.

Data sources: DGM: Geoportal Berlin/ATKIS ® DGM and Geoportal Berlin/ALKIS Gebäude; Infiltration based on Umweltatlas Berlin/Versiegelung 2021 and Umweltatlas Berlin/Bodenkundliche Kennwerte 2015; Land-use use from Geoportal Berlin/ALKIS Nutzung Flächen with Manning values from Klimaatlas NRW (2024)

**Figure 2.** Characteristics of the study area: Digital elevation model including buildings (left), infiltration rates (middle), and Manning's friction coefficients (right); the basic data used to create these maps are licensed under dl-de/by-2-0 (www.govdata.de/dl-de/by-2-0)

# 4 Results

195

## 4.1 Return levels

The relationship between intensity and duration for different event probabilities in the historical period of the climate simulation can be seen in Fig. 3. Due to the temporal resolution of the input data, the d-GEV could only be fitted using accumulation periods  $\geq 1$  hour (i.e. the region right of the vertical grey lines).

Heavy rain hazard maps are often based on an hourly 100-year event. For this probability and selected durations, Fig. 3a shows the return level from KOSTRA-DWD. The area-average of all cells located within the city limits of Berlin is denoted by a black cross, while the range of return levels from the individual cells is illustrated by the black horizontal line. The return levels from the historical simulation period match the station based KOSTRA-DWD values perfectly. For hydrodynamic modeling, an assumption must also be made about the temporal distribution of the hourly precipitation sum. A typically rainfall distribution used for design purposes and to generate pluvial flood maps is the so-called Euler-2 design rainfall (see Fig. 5).

To construct an Euler-2 precipitation time series, sub-hourly intensities down to 5 minutes need to be known. These durations are not resolved by the model simulation but can be extrapolated using Eq. 1. As the comparison with KOSTRA-DWD for the 5-minute duration (Fig. 3 b at 0.08 h) shows, this approach is problematic (Fig. 3a). In comparison to KOSTRA-DWD, the extrapolated sub-hourly intensities are overestimated. The reason for this is that due to the missing sub-hourly data the parameter  $\Theta$  describing the curvature of the IDF curves for sub-hourly durations, cannot be realistically fitted. The approach to overcome this problem is described in Sect. 4.3.

# 4.2 Climate change signal

210

A comparison between the three periods simulated with CCLM-CPS shows a clear increase in temperature with increasing GHG forcing for the study area (Fig. 4). In comparison with the mean temperature at the Berlin Dahlem station, the driving global model MIROC5 exhibits a positive bias, while the convection permitting model CCLM-CPS is around one Kelvin too cold during the reference period. The temperature trend in the scenario period is stronger in the global model than in the convection permitting model. In both models, the temperature variability beyond the trend decreases with increasing GHG concentrations.

The median of the hourly annual maximum rainfall intensities increases with increasing GHG concentrations (Fig. 1). The interannual variability is high and the events with the highest intensities occurred during the near future simulation period (simulation years 2045, 2051, 2059, and 2039). This is reflected in the statistical rainfall sums for probabilities associated with return periods beyond the length of the available data set (0.98 and 0.99), determined using the relationship described in 1. They increase compared to the historical reference period, but show a slight decrease when comparing the near future (2031-2060) to the far future (2071-2100; Fig. 3b). This is a result of the high variability in extreme precipitation that is superimposed on the overall increase (e.g. Hundhausen et al., 2024) and illustrates the need for long simulations for robust estimates. The difference between the two scenario periods is within the confidence intervals of the IDF fit.

In terms of percent the increase of the 1-hour, 100-year return level compared to the historical period is +46% and +42% for the periods 2031-2060 and 2071-2100, respectively.

## 230 4.3 Urban flooding

The following heavy rain events have been simulated with the coupled 2D-1D hydrodynamic model to study their impact on urban flooding in the selected area of Berlin:

- Historical 100a: 1-hour 100-year event of the CCLM-CPS simulation period 1971-2000 with a rainfall sum of 47.7 mm/h
   (-2% compared to the KOSTRA 100a event)
- Future 100a: 1-hour 100-year event of the CCLM-CPS simulation period 2031-2060 with a rainfall sum of 69.8 mm/h (+46% compared to the Historical 100a event)
  - Strongest: the strongest 1-hour event within the three 30-year periods of CCLM-CPS simulations with a rainfall sum of 106.7 mm/h (+123% compared to the Historical 100a event)

**Figure 3.** (a) Intensity-Duration-Frequency curves for the historical simulation. The colors distinguish 3 different non-exceedance probabilities (return periods). Shading denotes the 95% confidence interval. Black crosses mark the area-average values from KOSTRA for the 100-year return period. Black lines show the spread of the KOSTRA values for the 100-year return period over all Berlin grid cells. (b) Change in intensity between the historical period and the scenario period 2031-2060. The vertical grey line in (a) and (b) marks the temporal resolution of the model data (1-hour). Values for durations shorter than 1 hour are extrapolated. Please note the different scales.

Since the 100-year events for the near and far future simulations were very similar and the near future event even a bit stronger, 240 only the 100-year event of the near future period was chosen for the hydrodynamic simulations. For the construction of the

**Figure 4.** Temporal evolution of 2-m temperatures. Blue shows the time series of the driving model MIROC5 for the gridbox closest to central Berlin. Red is the time series for the 3 periods simulated with CCLM-CPS (mean over all grid boxes located within Berlin). As a reference the time mean for the period 1971-2000 from observation station Berlin is included in black. The gray shaded areas mark the 30-year periods downscaled with CCLM-CPS.

Euler-2 design rainfall, intensities for duration classes of less than one hour are needed. As already explained in section 4.1, the precipitation output from the climate simulations is one hour and extrapolating sub-hourly intensities from the modeled probability distribution leads to strong overestimations of sub-hourly intensities compared to values from KOSTRA. To overcome this problem, a Euler-2 time series is constructed by scaling the statistical hourly and sub-hourly 100-year return values from KOSTRA-DWD-2020. The scaling factor is applied to each time step and calculated as the ratio between the 1-hour rainfall sum of the respective event (as listed above) and the 1-hour rainfall sum of the 100-year event from KOSTRA-DWD-2020 (48.5 mm). The resulting scaling factors are 0.98 (47.7/48.5) for the Historical 100a event, 1.44 (69.8/48.5) for the Future 100a event, and 2.20 (106.7/48.5) for the Strongest event. For the Strongest event, the maximum intensity calculated with this approach is 45 mm in 5 min (9 mm/min). This is considered to be unrealistically high as the maximum 5-min rainfall measured at the station in Berlin-Dahlem during the period 1979 to 2023 was 12.3 mm. For an additional comparison with less extreme maximum intensities, all selected events were also simulated with a constant rainfall intensity. This also allows to analyze the influence of the temporal rainfall distribution. Fig. 5 shows the Euler-2 and constant rain events for the three aforementioned rainfall sums.

## 4.3.1 Effects of climate change

#### 255 Spatial distributions

Figure 6 shows the spatial distributions of simulated maximum water depths for the six simulated heavy rain events in the selected study area in Berlin. To enhance the visibility, not the whole, but a cropped extent of the model domain is depicted here. Water depths smaller 1 cm are set to transparent and are not visible. The top row illustrates the maximum water depths

**Figure 5.** Rainfall inputs for hydrodynamic flood simulations: hourly 100-year events for Berlin with temporal distribution after Euler-2 design rainfall and constant rain intensities; Euler-2 intensities are constant over each 5-min interval

for the three selected events with a temporal rainfall distribution according to the Euler-2 design rainfall, and the bottom row for the events with a constant rainfall intensity. The Gleimtunnel, a known local flooding hotspot (Berliner-Zeitung, 2024), is illustrated in the detail. Flooding areas and water depths significantly increase between the Historical 100a, the Future 100a, and the Strongest event as rainfall sums increase. The events with constant rainfall intensity generate smaller flooding areas and water depths than the Euler-2 rainfalls, which gets more pronounced with increasing rainfall sum. The flooding areas in the parks (visible through green color from the underlying OpenStreetMap) are much smaller in the case of constant than Euler-2 rainfall. This implies that during constant rainfall, less surface runoff is generated in the parks than during an Euler-2 rainfall, as the rainfall intensity rarely exceeds the infiltration rate of the soil. Even in the case of the Strongest event with constant intensity, there is almost no flooding in the parks.

Fig. 7 depicts the maximum surface runoff in terms of the unit discharge as product of water depth and flow velocity. According to Martínez-Gomariz et al. (2016), a value of  $0.22 \,\mathrm{m}^2/\mathrm{s}$  can be considered as threshold for the stability of pedestrians. In Fig. 7, orange to pink colors indicate surface runoff between 0.2 and  $0.4 \,\mathrm{m}^2/\mathrm{s}$ , exceeding the stability threshold, while yellow indicates a surface runoff between 0.02 and  $0.2 \,\mathrm{m}^2/\mathrm{s}$ , thus stable conditions for pedestrians.

It can be seen that the stability threshold is barely exceeded in the events with constant intensity. For the Euler-2 rainfalls, the Future 100a and the Strongest event lead to areas at the east entrance of the Gleimtunnel where the stability threshold is exceeded. Overall, the Euler-2 rainfalls lead to more surface runoff depicted by the yellow color on almost all roads, while the constant rainfalls show many more areas with surface runoff  $\leq 0.02 \text{ m}^2/\text{s}$ , which is set to transparent.

# Temporal developments at the Gleimtunnel

To get a better understanding of the flood development over time and the differences between the different events at one specific flooding hotspot, Fig. 8 depicts the temporal developments of water depth (left) and surface runoff (middle) as maximum

**Figure 6.** Maximum water depths (m) for different rainfall sums (Historical 100a, Future 100a, and Strongest event) and temporal distributions (Euler-2 and constant) (values smaller 0.01 m are set to transparent and are not shown). Background:

values over one longitudinal section through the Gleimtunnel. For both temporal rainfall distributions, constant and Euler-2, the peak water depth significantly increases with increasing rainfall. The differences in the magnitude of peak water depth between constant and Euler-2 rainfall distributions are much larger for the Strongest event than for the Historical 100a and Future 100a events. The increase in peak water depth between the Historical and the Future 100a event is 51% for Euler-2 and 47% for constant rainfall, and between the Historical 100a and Strongest event 137% for Euler-2 and 109% for constant rainfall. The time to peak mainly depends on the rainfall distribution and not on the rainfall sum. The constant rainfalls generate peak water depths around half an hour later than the Euler-2 rainfalls, which is shortly after the rainfall stopped. The temporal developments of surface runoff in the middle panel of Fig. 8 show that the Strongest event leads to an earlier start and much higher magnitudes of surface runoff than the Historical 100a and Future 100a events, while in case of Euler-2 rainfalls, the main peak of the Strongest event occurs later than the ones of the Historical 100a and Future 100a events. The Historical 100a event with constant rainfall does not generate a pronounced peak runoff and shows overall relatively small surface runoff values of maximum 0.02 m<sup>2</sup>/s.

As a coupled model of the flow on the surface and in the subsurface drainage system is used, the effects of different rainfall amounts and temporal developments in the drainage system can be assessed as well. Fig. 8 (right) illustrates exemplarily the results of the outflow at one model outfall, which reflects combined sewer overflow (CSO) into the nearby river. For the simulated period, the CSO average flow rate and overall volume increase from the Historical 100a to the Future 100a event by

Figure 7. Maximum surface runoff ( $m^2/s$ ) for different rainfall sums (Historical 100a, Future 100a, and Strongest event) and temporal distributions (Euler-2 and constant) (values 0.02  $m^2/s$  are set to transparent and are not shown)

33% for Euler-2 rainfall and 27% for constant rainfall, and from the Historical 100a to the Strongest event by 74% for Euler-2 and 68% for constant rainfall. As depicted in Fig. 8, the flow has not stopped yet at the end of the simulation time, so the differences in the final overall volumes might be a bit different than the numbers given above. In the case of constant rainfall, CSO starts later and has later peaks, but with almost the same magnitudes. In the following, effects of infiltration, the drainage system, and the retention of rainwater from roofs are presented for different heavy rain events. For simplicity, only results with Euler-2 rainfalls are presented.

## 4.3.2 Effects of selected mitigation measures

The setup from the previous section - namely including infiltration and the subsurface drainage system, and without retention roofs - is used as the reference case representing the current state of the study area while neglecting the few existing green roofs. In the following, three different setups are compared to the reference case for different rainfall events: 1) simulations without infiltration to indirectly estimate the negative impacts in case of further surface sealing or on the other hand the effectiveness of large-scale surface desealing, 2) simulations without the drainage system to quantify the effectiveness of the actual subsurface grey infrastructure for extreme rainfalls under current and future climate, and 3) with all roof surfaces as retention roofs representing one example of blue-green infrastructure through the idealization that all roofs can capture the whole rainfall, even in the case of the Strongest event. These simulations are not intended to represent realistic scenarios, but rather to explore the

**Figure 8.** Temporal development of max. water depth (left) and surface runoff (middle) at the Gleimtunnel, and combined sewer overflow into the nearby river at one model outfall (right)

range of possible system behaviors and assess the sensitivity of flooding dynamics to infiltration, drainage, and retention roofs. For example, the case of full retention of the rain on all roof surfaces can be seen as a best-case scenario. While the complete retention during all considered heavy rain events could hardly be realized by modifying all roofs in the study area to become green or retention roofs, it could - in theory - be achieved with combinations of green and/or retention roofs and cisterns for all buildings in that area. The stored water could also help conserve drinking water by being used for non-potable purposes, such as toilet flushing or irrigation. A realistic scenario of considering only suitable roofs for retention roofs, for example based on their inclination, is beyond the scope of this study.

#### **Spatial distributions**

Figure 9 depicts the spatial distributions of the differences in maximum water depths between the simulations without and with infiltration, drainage, and retention roofs, respectively. The left panels represent the Historical 100 event, and the right panels the Strongest event, both for Euler-2 rainfalls. Blue colors show the reductions while orange to red colors indicate increases due to the considered settings, which occurs only in the middle panels (drainage) and is further highlighted by black circles. The top panels illustrate the differences between the simulations without and with infiltration. Reductions between 1 and 10 cm occur in large parts of the model domain, while at flooding hotspots, such as the Gleimtunnel, reductions between 10 and 30 cm have been simulated. At a few locations, the simulation results show reductions of 30 cm and more.

The middle panels display the differences between the simulations without and with consideration of the drainage system. The results illustrate, that the drainage system effectively reduces flooding areas and water depths specifically at local hotspots, even in the case of the Strongest event. Nevertheless, high water depths still occur in large parts of the model domain as shown

in the results for the reference case with drainage in section 4.3.1. For the Historical 100a event as well as the Strongest event, reductions of more than 1 m in the area of the Gleimtunnel and between 0.5 and 1 m at another local flooding hotspot are reached due to the drainage system. At several other roads, reductions between 0.1 and 0.5 m occur. The areas in orange to red color (within the black circle) indicate overflow from the drainage system to the surface. This is simulated only at one location at railway tracks northwest of the Gleimtunnel. While this proofs the functioning of the bi-directional coupling of the surface and the drainage model, the plausibility of overflow at this location would need to be further checked, as it might be generated due to missing information on additional drainage infrastructure from the train company, which is not included in the used drainage model.

The bottom panels illustrate the spatial distribution of differences in maximum water depths for the simulations without and with full retention of rain on all roof surfaces. In large parts of the model domain, the retention roofs lead to reductions of up to 10 cm for both events. In some backyards, reductions of more than 40 cm are reached, but these details are not easily visible at the shown scale. Furthermore, the retention roofs lead to significant reductions of the flooded area in and around the Gleimtunnel, and the maximum water depths here are reduced by up to 30 cm in the Historical 100a event and up to 40 cm in the Strongest event.

# Temporal developments at the Gleimtunnel

Figure 10 shows the temporal development of the maximum water depth (left) and surface runoff (right) at the Gleimtunnel for the different scenarios. In the top panel, the results with and without infiltration are depicted. The peak water depth of the Historical 100a, Future 100a, and the Strongest event increased by 33%, 24%, and 18% respectively, if infiltration is not considered. While the peak runoff of the Future 100a and Strongest events increased by 9% and 21%, respectively, it is nearly the same for the Historical 100a event. The surface runoff volume at the Gleimtunnel increased in all events. In addition, the combined sewer overflow volume increased by 19-30%.

The middle panels compare the results without and with consideration of the drainage system. On the left, the temporal developments of maximum water depths illustrate the effectiveness of the drainage system. In the case with drainage, the water depth decrease over time in all rainfall scenarios, while in the case without drainage, water depths are still rising until the end of the simulation time of two hours, while the rainfall stopped already after one hour. The maximum water depth until the end of the simulation time is 170% (Historical 100a), 154% (Future 100a), and 110% (Strongest event) higher without drainage compared to the simulations with drainage. Even though the effectiveness of the drainage system decreases with increasing rain intensity, its tremendous effect on flood reduction is still given for the Strongest event. The right middle panel shows the effects of drainage on the surface runoff. While for the Historical 100a and the Future 100a events, both peak and volume of surface runoff are higher without the drainage system, the hydrograph shape for the Strongest event appears differently, showing a much faster decline than in the case with drainage. This can be explained by the (earlier) generation of a larger and deeper flooding in and around the tunnel reducing the flow velocity inside the longitudinal section that was considered for the temporal developments at the Gleimtunnel.

**Figure 9.** Spatial distributions of differences in maximum water depths for Euler-2 rainfalls between: without and with infiltration (top), without and with consideration of the drainage system (middle), and without and with full retention on all roofs (bottom); negative differences (orange to red colors, within black circle) indicate increases in maximum water depth due to overflows from the drainage system to the surface

The bottom panels illustrate the effects of retention roofs on maximum water depths and surface runoff at the Gleimtunnel. The peak water depth is reduced by 22% (Historical 100a), 23% (Future 100a), and 24% (Strongest event), respectively,

showing very similar numbers and a small tendency to higher effectiveness with increasing rainfall intensity. Full retention was considered for all three events, but in the case of the Strongest event with 106.7 mm/h, it has to be pointed out that a full retention of more than 100 l/m² is less realistic than the full retention of about 50 or 70 l/m² as considered in the other two events. Overall, as noted earlier, treating all roofs as retention roofs represents an idealized best-case scenario. The temporal developments of surface runoff depicted on the right illustrate how much the surface runoff could be reduced through retention roofs, especially in the case of the Strongest event. While the simulation without retention results in a clear exceedance of the stability threshold of 0.22 m²/s after Martínez-Gomariz et al. (2016), the simulation with retention shows that the peak runoff could be reduced below the stability threshold. In the spatial distributions (which are not shown) it gets clear, that the threshold is still exceeded, but in a smaller area, which was not included in the longitudinal section considered for the extraction of temporal developments at the Gleimtunnel. Still, it emphasizes that the retention roofs in the surroundings not only reduce the water depths at local flooding hotpots but also dangerous flow conditions for pedestrian and possibly cars, which is not considered here in more detail. The retention roofs reduced the combined sewer overflow volume by about 15-20%.

# 4.3.3 Flood volumes

To compare the effects of rainfall sum and temporal distribution, infiltration, drainage, and retention roofs on the flooding in the whole domain, the flood volume at different time steps is calculated as sum of the water depths in all cells multiplied by the cell area. Here, the domain has been cropped to exclude areas at the boundaries, for which the drainage system was not included. Results at different timesteps were compared, selecting 30 minutes to represent the peak time of rainfall and water depth at the Gleimtunnel and 120 minutes representing the end of the simulation time. Table 2 lists the relative differences for all simulations compared to the respective Euler-2 rainfall. It also includes the values for three additional simulations addressing spatial resolution, additional underground storage, and the interplay between infiltration and drainage. First, the results of the simulations presented in the previous sections will be discussed, followed by the results of the additional simulations.

## Effects from rainfall sum and temporal distribution, infiltration, drainage, and retention roofs

A first conclusion is that, under Euler-2 rainfall distributions, increases in the rainfall sum of 46% (Future 100a) and 123% (Strongest event) lead to increases in flood volume of 66% (Future 100a) and 189% (Strongest event) after 30 minutes, and 66% (Future) and 219% (Strongest) after 60 minutes, compared to the Historical 100a event. The results reveal a non-proportional relationship between rainfall sum and flood volume, with flood volumes at both time steps increasing more strongly than the rainfall sum, particularly for the Strongest event after 120 minutes.

When comparing Euler-2 to constant rainfall, constant rainfall produces 51% (Historical 100a), and 54% (Future 100a and Strongest event) smaller flood volumes after 30 minutes than the corresponding Euler-2 event. By the end of the rainfall event, however, the reductions are much smaller with 7% (Historical 100a), 13% (Future 100a), and 15% (Strongest event). These significant differences highlight the importance of temporal rainfall distribution particularly for peak values and even more for increasing rainfall sums, and the need of sub-hourly rainfall intensities.

**Figure 10.** Temporal development of water depth and surface runoff within the Gleimtunnel without and with infiltration (top), the drainage system (middle), full retention roofs (top) - for Euler-2 rainfalls, maximum values along a longitudinal section through the Gleimtunnel; solid lines represent the reference case (with drainage and infiltration, without retention roofs)

Simulations without infiltration show increases in flood volume by 83% (Historical 100a), 60% (Future 100a), and 39% (Strongest event) after 30 minutes, which increase to 251% (Historical 100a), 201% Future 100a), and 132% (strongest event) after 120 minutes. Thus, even in such a densely urbanized area (51% sealed by buildings and roads, with an average sealing degree of 26% for unbuilt surfaces), infiltration strongly influences surface flooding, although its relative importance decreases as rainfall sums increase.

The combined effects of temporal rainfall distribution and infiltration, as well as time-dependent infiltration approaches, could be further explored but are beyond the scope of this study. The effect of the drainage system on the overall flood volume is reflected by changes of 23% and 75% (Historical 100a), 20% and 67% (Future 100a), and 18% and 52% (Strongest event) after 30 and 120 minutes, respectively. This indicates that the drainage system has a much stronger influence on the flood volume towards the end of the simulation than during the peak, with decreasing effectiveness with increasing rainfall sum. As shown in the previous section, the drainage system is particularly effective in reducing the highest water depths at local hotspots. Retention roofs reduce the flood volume by 39% (Historical and Future 100a) and 37% (Strongest event) after 30 minutes. After 120 minutes, the reductions are 32% (Historical), 36% (Future), and 39% (Strongest). Since the consideration of retention roofs decreases the same percentage of the total rainfall volume in all scenarios, their effect on flood volume is very similar across events. For the Historical 100a event, the influence on flood volume after 30 minutes can be ranked as follows: 1) Infiltration (83%), 2) Temporal rainfall distribution (51%), 3) Retention roofs (39%), and 4) Drainage (23%). For the Strongest event, the ranking is: 1) Temporal rainfall distribution (54%), 2) Infiltration (39%), 3) Retention roofs (36%), and 4) Drainage (18%). After 120 minutes, the effectiveness of the drainage system is enhanced, reflected by increases of 75% (Historical 100a) and 52% (Strongest event) between simulations with and without drainage, moving it to rank 2. The effect of infiltration becomes more pronounced at the end of the simulation as more water can infiltrate over the longer time span. This leads to increases by 251% for the Historical 100a and 132% for the Strongest event between simulations with and without infiltration, ranking it 1 for both events. The temporal rainfall distribution has a smaller impact on the final flood volume, showing a 7% reduction for the Historical 100a and a 15% reduction for the Strongest event when comparing constant to Euler-2 rainfall, moving it to rank 4. The effect of retention roofs remains similar, maintaining rank 3. In conclusion, all three events show the same ranking of influencing factors on the flood volume after 120 minutes: 1) Infiltration, 2) Drainage, 3) Retention roofs, and 4) Temporal rainfall distribution.

#### Effects of spatial resolution, additional storage, and the interplay of infiltration and drainage

For the Historical 100a event, one simulation with a spatial resolution of 2 m instead of 5 m was carried out. The overall flood volume increased by 1% after 30 as well as 120 minutes, which can be considered as a relatively small effect of the finer model resolution. Also, the temporal development of maximum water depth at the Gleimtunnel is almost the same (not shown here). The computation time increased by a factor of 3.3 from 2.4 hours (5 m resolution) to 7.8 hours (2 m resolution) while using 48 CPU cores (Intel Cascade Lake Platinum 9242) for both simulations, showing a non-linear scaling behavior. Furthermore, one simulation without infiltration and drainage system was carried out, showing a 17% higher flood volume increase after 120 minutes than if the increases of the simulations without either infiltration or the drainage system are summed

up (271% compared to 40% + 214% = 254%). This reflects the interaction of both processes - infiltration and drainage through the underground pipe system. Further additional simulations for all three heavy rain events were carried out without the underground Mauerpark storage, which was completed in 2020 and provides an additional storage volume of 7500 m³ in the drainage system. Comparing the simulation results without the Mauerpark storage to the reference simulations with Mauerpark storage showed a difference of 1-2% in the flood volume after 120 minutes for all three rainfall events. The spatial distributions of maximum water depths (not shown here) revealed that the Mauerpark storage primarily reduced water depths near the Gleimtunnel, close to the storage location. Additionally, it decreased surface overflow from the drainage system at at more distant locations. For the Historical 100a and the Future 100a events, the percentage reductions in maximum water depth at the Gleimtunnel are 16% and 9%, respectively, and in the combined sewer overflow volume 3% and 1%, respectively, indicating a decreasing effect with increasing rainfall intensity.

Table 2. Relative differences in flood volumes after 30 and 120 minutes

|                                   | After 30 minutes                           |        |           | After 120 minutes                               |        |           |  |
|-----------------------------------|--------------------------------------------|--------|-----------|-------------------------------------------------|--------|-----------|--|
|                                   | Historical                                 | Future | Strongest | Historical                                      | Future | Strongest |  |
| Case                              | Relative difference to Historical Euler-2  |        |           | Relative difference to Historical Euler-2       |        |           |  |
| Euler-2                           | -                                          | +66%   | +189%     | -                                               | +66%   | +219%     |  |
|                                   | Relative differences to respective Euler-2 |        |           | Relative differences in % to respective Euler-2 |        |           |  |
| Constant                          | -51%                                       | -54%   | -54%      | -7%                                             | -13%   | -15%      |  |
| Without Infiltration              | +83%                                       | +60%   | +39%      | +251%                                           | +201%  | +132%     |  |
| Without Drainage                  | +23%                                       | +20%   | +18%      | +75%                                            | +67%   | +52%      |  |
| With Retention Roofs              | -39%                                       | -38%   | -36%      | -32%                                            | -36%   | -39%      |  |
| Cellsize 2 m                      | +2%                                        | -      | -         | +1%                                             | -      | -         |  |
| Without Infiltration and Drainage | +110%                                      | -      | -         | +357%                                           | -      | -         |  |
| Without Mauerpark Storage         | 0%                                         | 1%     | 1%        | +3%                                             | +4%    | +3%       |  |

# 4.3.4 Sensitivity of infiltration settings and effects of further desealing

Since infiltration is subject to uncertainties, additional simulations were carried out to test the effects of different infiltration settings by varying (1) the percentage of surface sealing and (2) the infiltration capacity, represented by the saturated hydraulic conductivity in the upper 1 m of soil. In section 4.3.2, streets were assumed to be fully impermeable, whereas here the sealing degree was applied both to block and street areas using data from Umweltatlas Berlin (2022). As the hydraulic conductivities for streets are not included in the public datasets, they were spatially interpolated. "Street areas" also include parking lots and sidewalks. Because streets act as the main flow paths in urban topography, we additionally tested the effect of a 10% reduction in sealing of street areas.

Furthermore, the saturated hydraulic conductivities from two publicly available datasets were compared, which are called kf 2015 (Umweltatlas Berlin, 2019) and kf 2020 (Umweltatlas Berlin, 2024). While they are based on data collections from 2015 and 2020, they were published in 2019 and 2024, respectively. Table 3 summarizes the resulting reductions (relative to the case without infiltration) in maximum water depth at the Gleimtunnel, CSO volume at the selected outfall, and flood volume after 120 minutes for the Historical 100a event. Results show that kf 2020 leads to smaller reductions than kf 2015, but both datasets still produce substantial reductions compared to the case without infiltration. As expected, the influence of hydraulic conductivity decreases with higher sealing degrees. The +10% unsealing scenario further reduces CSO volume by 11% (kf 2020) and 15% (kf 2015), maximum water depth at the Gleimtunnel by 5% (kf 2020) and 4% (kf 2015), and flood volume after 120 minutes by 4% (kf 2020) and 5% (kf 2015).

**Table 3.** Percentage reductions of CSO volume, maximum depth at the Gleimtunnel, and flood volume after 120 minutes compared to the case without infiltration (all simulations for the Historical 100a event)

| Case                                           | CSO volume | Max. depth Gleimtunnel | Flood volume (120 min) |
|------------------------------------------------|------------|------------------------|------------------------|
| kf 2015 only block areas partly unsealed       | 23%        | 25%                    | 69%                    |
| kf 2020 only block areas partly unsealed       | 23%        | 21%                    | 62%                    |
| kf 2015 street and block areas partly unsealed | 34%        | 32%                    | 73%                    |
| kf 2020 street and block areas partly unsealed | 26%        | 25%                    | 64%                    |
| kf 2015 plus 10% unsealed on street areas      | 49%        | 36%                    | 78%                    |
| kf 2020 plus 10% unsealed on street areas      | 37%        | 30%                    | 68%                    |

## 4.4 Discussion

In this study the effect of climate change on extreme precipitation probability is studied and the resulting urban flooding is quantified. For our study region Berlin only one global and regional climate model combination (MIROC5-CCLM) with convection permitting resolution was available at the time of the study. The analysis is based on the three available 30-year simulation periods forced with observed GHG concentrations (1971-2000) and RCP8.5 conditions for the periods 2031-2060 and 2071-2100. For Berlin the simulations show an increase in extreme precipitation frequency and intensity during the climate scenario periods compared to the historical period for short duration events that are relevant for pluvial flooding. Such an increase is also reported by other studies analyzing other regions in Germany with different regional model simulations at comparable resolution (Meredith et al., 2019; Hundhausen et al., 2024). While the median of the annual maximum hourly precipitation events in the MIROC5-CCLM simulations increases with increasing greenhouse gas concentrations, the most extreme events over Berlin are simulated during the near future rather than the far future period. As a consequence, the hourly return levels for the 100-year event, which we use as input for the hydrodynamic simulations, are highest during the near future period. This is unexpected as other studies suggest a steady increase of extreme precipitation probability which is most pronounced for high return periods and short durations (e.g. shown for Southern Germany in Hundhausen et al., 2024). The

result can be explained by the high variability in extreme precipitation that was found to increase with increasing greenhouse gas warming (Hundhausen et al., 2024). To obtain more robust results an ensemble of climate scenario simulations is required. In order to address the uncertainties associated with the climate scenario, the ensemble should comprise simulations forced with different greenhouse gas concentration pathways. Such an ensemble is currently produced under the NUKLEUS project framework and should be soon available for future studies (BMBF, 2024).

To quantify the effects of changes in heavy rainfall on urban flooding, hydrodynamic simulations are needed, preferably by coupling flow processes on the surface and in the drainage system. Such simulations always include some simplifications and assumptions. In this study, flow processes on the surface and in the subsurface drainage system have been bi-directionally coupled. As shown in the results, infiltration had a significant effect on the flooding, even though the study area is highly urbanized, emphasizing the importance of urban green areas and possible desealing of currently sealed surfaces. But the spatial extent of such measures would need to be large to have significant effects. Uncertainties in both, soil and surface sealing data, have been addressed in a small sensitivity study. One simplification of the model is that direct drainage from roofs into the drainage system was not included. Instead, runoff from rainfall on the roofs flows to adjacent areas. For streets, water can enter the subsurface drainage system via street inlets, which provides a reasonable approximation of actual roof drainage directly into the system, noting that this is effective for only part of the rainfall volume. Some runoff also flows into backyards, where no connection to the subsurface drainage system is represented in the model. This may lead to a slight overestimation of surface flooding, as some of the water would likely be drained into the subsurface system rather than reaching the backyard. In contrast, this implies a potential underestimation of water in the subsurface drainage system and, consequently, in the combined sewer overflow. Furthermore, local details such as culverts or building passages were not explicitly considered in this study.

In the current study, retention roofs are modeled using a simplified approach. In contrast, Neumann et al. (2024) applied Low Impact Development (LID) elements in the Storm Water Management Model (SWMM) within a 2D-1D surface runoff model for a different study area also in Berlin. In their study, the roof areas are not modeled as 2D surfaces, but as subcatchments connected to the nearest manhole. Additionally, retention roofs are defined in greater detail, using a list of parameters and multiple layers. Their findings demonstrated that retention roofs could fully retain rainfall from a 1-hour event with an intensity of 100 mm/h. Therefore, our simplified assumption of full retention is consistent with their results, even for the Strongest event of 106.7 mm/h.

Comparing the effects of infiltration, drainage system, retention roofs, and the additional storage below the Mauerpark on the maximum water depth in the local flooding hotspot Gleimtunnel gives the following ranking for the Historical 100a event: 1) Drainage (170%), 2) Infiltration (33%), 3) Retention roofs (22%), and 4) Mauerpark storage (16%). The effects on the overall flood volume after 30 minutes are ranked differently: 1) Infiltration (83%), 2) Retention roofs (39%), 3) Drainage (23%), and 4) Mauerpark storage (less than 1%). At the end of the simulation time, the reduction due to drainage increases to 75%. The drainage system is particularly efficient in reducing flooding at hotspots and to later timesteps than during the peak, the Mauerpark storage contributes to some extent to mitigating flooding specifically at Gleimtunnel, and the presented scenarios of infiltration and retention roofs across the entire catchment have distributed effects, while if realized in large spatial extent as in the given scenarios, they also contribute significantly to reductions at local hotspots. All effects decrease with increasing

rainfall amount, except for the retention roofs, which by definition were able to retain the total rainfall for each of the three events in this study. Additional simulations of different unsealing degrees on street areas showed significant effects on the CSO volume, maximum depth at the Gleimtunnel, and the overall flood volume.

In Germany and several other European countries urban sewage systems are designed based on design rainfalls of Euler-2 type, where the type describes the temporal evolution of the event (hyetograph). Maps for pluvial flood risk in Berlin are based on Euler-2 type statistical 100-year hourly events. Two different types of design rainfalls (Euler-2 and constant) are compared in this study. The temporal evolution of the event influences the impacts. This has also been reported by Paton et al. (2024). While Paton et al. (2024) found that in Germany the severity of observed short extreme rainstorms often exceed the severity of the Euler-2 design rainfall events, for Poland Wartalska et al. (2020) found that Euler-2 design rainfalls have the tendency to be significantly larger than historic events. Both studies indicate that it would be desirable to use more realistic hyetographs than design rainfalls for the construction of risk maps.

The findings generally apply also to other cities with similar hydrological characteristics. The increase in extreme rainfall intensities has been already observed in many regions of the world, and according to climate projections significant increases in the maximum daily rainfall intensity are expected for almost all land surfaces, while the percentages of increase depend on the emission scenario (IPCC, 2021). The effects of infiltration and the drainage system are expected to also be similar depending on the design of the subsurface drainage system, soils, surface sealing conditions, and topography. The approach can be applied to any other city, while often the availability of high-quality and high-resolution data will be a limiting factor. Particularly, data or a model of the subsurface drainage system might often not be available or accessible. Reasonable drainage models could be generated by using AI and available data, e.g. on topography, roads, manholes, and sealed surfaces (Döring and Neuweiler, 2019). High-resolution DEMs based on LiDAR data as well as more detailed land use maps are getting more often available. Data on hydraulic soil properties are often a limiting factor, but associated uncertainties can be addressed through sensitivity analyses.

Different studies showed similar effects of climate change and blue–green measures on urban flooding while using (partly) different methods. For example, Yang et al. (2024) demonstrated that climate change substantially increases the severity of urban flooding, particularly during short-duration extreme events. In contrast to our event-based, high-resolution 2D–1D hydrodynamic simulations, which used climate projections to derive extreme rainfall statistics and investigate spatial–temporal impacts, they coupled downscaled GCM scenarios with SWMM for long-term simulations of trends in annual runoff, flood volume, and the frequency of daily heavy rainfall in Dresden (Germany). Similarly, Li et al. (2024) combined CMIP6 climate projections with SWMM for a 6 km<sup>2</sup> catchment in Haining City (China) and found that under SSP5–8.5, urban flood volumes from overflowing manholes could increase by more than 37% by the end of the century. Both studies, however, did not include 2D surface flow simulations, which enable high-resolution spatial analysis of flood depths and hotspots. Regarding blue–green measures, Liu et al. (2024) developed a dynamic framework coupling SWMM with GIS spatial analysis for Zhengzhou (China). They found that LID measures reduce runoff and inundation, but their effectiveness decreases with increasing rainfall intensity and is strongest under uniform rainfall patterns. This is consistent with our findings of reduced effectiveness under more extreme rainfall and a strong influence of temporal rainfall distribution. Similarly, Hua et al. (2020) showed for Chaohu City

(China) that LID measures effectively reduce inundation, with higher performance under smaller storm magnitudes. Their model combined rainfall—runoff and 1D pipe flow with 2D overland flow, but the latter was only applied for surcharged runoff. By contrast, our study directly applies rainfall to the 2D grid and resolves surface—drainage interactions bi-directionally, capturing flow both before entering the drainage system and during overflow events.

#### 5 Conclusions

560

565

- In this study the effect of climate change on extreme precipitation probability is studied and the resulting urban flooding is quantified. It has been taken care to include all relevant processes in the hydrodynamical simulations and represent them based on given data and knowledge. This includes a fully dynamic 2D robust shallow water model coupled to a 1D subsurface drainage model provided by the city's water company, and spatially distributed friction and infiltration parameter depending on the land use, soils, and degree of sealing. Our most important results are:
- Under climate change conditions the frequency and intensity of extreme precipitation events is expected to increase. The climate scenario simulations analyzed in this study for Berlin, Germany, predict a 46% increase in the intensity of a statistical 1-hour 100-year event in Berlin under RCP8.5 climate scenario conditions until the year 2060.
  - The relationship between flooding and rainfall is non-linear. It depends on the surface conditions, infiltration and drainage capacities, and the temporal evolution of the precipitation event. The same amount of rain but with temporally higher precipitation intensities leads to more severe flooding in terms of water depth, surface runoff and combined sewer overflow.
  - Mitigation measures should be combined. The subsurface drainage system is most effective in reducing water depth at flooding hotspots, while infiltration is most effective in reducing the overall flood volume. Retention roofs support these measures and can reduce water depths and flood volumes significantly. Infiltration and retention also significantly reduce the combined sewer overflow volume.

The study is associated with a number of uncertainties, the most important being:

- The choice of the RCP8.5 climate scenario for the rainfall analysis and the fact that only one global-regional model combination was available for the study. This limitation affects the estimation of the strength of the precipitation change, that is at the upper range of changes reported by Hundhausen et al. (2024) for the area average in southern Germany.
- The highest uncertainties for flood calculations can be assumed to be associated with infiltration, which depends both. on soil and surface sealing data. A sensitivity study on sealing degree and saturated hydraulic conductivity shows that uncertainties in infiltration translate to roughly 10% variation in the analyzed flood characteristics.

The fact that extreme precipitation intensifies under global warming conditions can be regarded as undisputed and has been recognized by a large number of studies (IPCC, 2021; Fowler et al., 2021). Irrespective of the various uncertainties associated

with this analysis, the investigation shows very clearly that urban flood risk maps must take climate change into account in order to adequately contribute to the protection of the population. Risk maps based on past observations with the assumption of a stationary climate will be soon outdated.

In this study we were able to demonstrate how hydrodynamic simulations can help to compare the effectiveness of different mitigation measures within an urban drainage catchment. A combination of measures will probably be needed to compensate for the effect of climate change. The plans of the Senate of Berlin to foster a transformation of the city into a "sponge city" by strengthening the role of green and blue infrastructure is a step towards a mitigation of the adverse effects but a more systematic and faster implementation is required.

Future work might encompass to assess further mitigation strategies. We found that replacing all customary roofs with retention roof is not sufficient to fully compensate the effect of climate change under the investigated scenario. Additional or alternative methods could be the inclusion of retention measures, i.e. rain gardens or other retention areas. Realistic scenarios of blue-green infrastructure and surface desealing could be elaborated in further modeling studies. Overall, multi-functional solutions combining several purposes are to be targeted. The sensitivity analysis explored the importance of infiltration on the impacts as well as associated uncertainties.

In this study, we focus on the Gleimtunnel, as it has a long history of flooding. However, the tunnel is of minor importance for Berlin's traffic network, and flooding there primarily causes local traffic disruptions and economic damages to car owners. Future studies could focus on flooding hotspots with high vulnerability, such as areas with critical infrastructure, schools, kindergartens, or social facilities located in low-lying areas.

Code and data availability. Station data for Berlin data is available from the open data server of the DWD: https://opendata.dwd.de/climate\_environment/CDC/observations\_germany/climate/hourly/precipitation/historical/. The climate simulations are available from https://esgf. dwd.de/projects/dwd-cps (Haller et al., 2022a, b). KOSTRA design rainfalls can be downloaded from https://opendata.dwd.de/climate\_environment/CDC/grids\_germany/return\_periods/precipitation/KOSTRA/. Surface data for Berlin is available from Geoportal Berlin under dl-de/by-2-0 licence (www.govdata.de/dl-de/by-2-0). The code for hms<sup>++</sup> is available on github (https://git.tu-berlin.de/wahyd/hmspp/hms

Author contributions. FT: conceptualisation, methodology, formal analysis, investigation, data curation, writing (original draft as well as review and editing), visualisation. KMN: conceptualisation, methodology, formal analysis, investigation, data curation, writing (original draft as well as review and editing), visualisation. LS: methodology, software, review. YZ: methodology, software, review. UU: supervision, review, project administration, funding acquisition. RH: supervision, review, project administration, funding acquisition.

Competing interests. U. Ulbrich is an editor for NHESS and K. Nissen a guest editor

Acknowledgements. Funding was received through the Einstein Research Unit 'Climate and Water under Change' from the Einstein Foundation Berlin and Berlin University Alliance, Grant number ERU-2020-609. This work used resources of the Deutsches Klimarechenzentrum (DKRZ) granted by its Scientific Steering Committee (WLA) under project ID 1229. The authors gratefully acknowledge the computing time made available to them on the high-performance computer "Lise" at the NHR Center NHR@ZIB. This center is jointly supported by the Federal Ministry of Education and Research and the state governments participating in the NHR (www.nhr-verein.de/unsere-partner). We would like to acknowledge Berliner Wasserbetriebe for providing their drainage model. The AI language model ChatGPT and the AI-powered writing assistant Grammarly were used to assist with text optimization and refinement in parts of this manuscript.

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
