# Peer review of "Extreme precipitation and flooding in Berlin under climate change and effects of selected grey and blue-green measures"

_EGUsphere, 2025_

## Referee Comment (RC2)

**Extreme precipitation and flooding in Berlin under climate change and effects of selected grey and blue-green measures**

Franziska Tügel[1,3,4], Katrin M. Nissen[2], Lennart Steffen[1], Yangwei Zhang[1], Uwe Ulbrich[2], and Reinhard Hinkelmann[1]

[revised manuscript text omitted]

165   0.12 and 2 m. The Manning's friction coefficient of all conduits is 0.013 s/m$^{1/3}$, the entry loss coefficient 0.15 and the exit loss coefficient 0.015.

For the surface flow model, spatially distributed friction and infiltration values were generated based on soil data, land use, and sealing degrees. Manning's friction coefficients for the different land use types were selected from recommendations given

in Klimaatlas NRW (2024). Infiltration was considered with spatially distributed constant infiltration rates calculated based
170   on the saturated hydraulic conductivity from the first 1 m of soil provided in the dataset Umweltatlas Berlin/Bodenkundliche
Kennwerte 2015 multiplied with the unsealing degree calculated from sealing degrees of the unbuilt surfaces on a block area
level provided in the dataset Umweltatlas Berlin/Versiegelung 2021. The average sealing degree of the unbuilt surfaces is about
26%. Buildings and roads were considered as completely sealed with an infiltration rate of zero, which is the case for 51% of
the considered study area consisting of 35% roof and 16% road surfaces. The digital elevation model was modified to cut free

[revised manuscript text omitted]
 address this, the selected approach involves using the sub-hourly Euler-2 distribution from the 100-year event according to KOSTRA-DWD-2020. Each 5-minute intensity value is scaled by a change factor, calculated as the ratio between the simulated 1-hour rainfall sum of the respective event and the 1-hour rainfall sum of the 100-year event from KOSTRA-DWD-2020 (48.5 mm).

[Figure]

[Figure]

The resulting change factors are 0.98 (47.7/48.5) for the Historical 100a event, 1.44 (69.8/48.5) for the Future 100a event,
and 2.20 (106.7/48.5) for the Strongest event. For the Strongest event, the maximum intensity calculated with this approach is
almost 9 mm/min, which would mean 45 mm in 5 min (as the used time step size for the Euler-2 construction is 5 min). This is
considered to be unrealistically high (the maximum 5-min rainfall measured at the station in Berlin-Dahlem during the period
1979 to 2023 was 12.3 mm), but the results from similar temporal rainfall distributions are easier to compare to each other. To
have a further comparison, all selected events were also simulated with a constant rainfall intensity. The results can then also
be compared concerning the influence of the temporal rainfall distribution. Fig. 5 shows the Euler-2 and constant rain events
for the three aforementioned rainfall sums.

[revised manuscript text omitted]

As a coupled model of the flow on the surface and in the subsurface drainage system is used, the effects of different rainfall amounts and temporal developments in the drainage system can be assessed as well. Fig. 9 presents exemplarily the results of the outflow at one model outfall, which reflects combined sewer overflow (CSO) into the nearby river. For the simulated period, the CSO average flow rate and overall volume increase from the Historical 100a to the Future 100a event by 33% for Euler-2 rainfall and 27% for constant rainfall, and from the Historical 100a to the Strongest event by 74% for Euler-2 and 68% for constant rainfall. As depicted in Fig. 9, the flow has not stopped yet at the end of the simulation time, so the differences in the final overall volumes might be a bit different than the numbers given above. In the case of constant rainfall, CSO starts later and has later peaks, but with almost the same magnitudes. In the following, effects of infiltration, the drainage system, and the retention of rainwater from roofs are presented for different heavy rain events. For simplicity, only results with Euler-2 rainfalls are presented.

**4.3.2 Effects of selected mitigation measures**

The setup from the previous section - namely with consideration of infiltration and the subsurface drainage system, and without retention roofs - is considered as the reference case. This can be considered as current state of the considered study area, neglecting the few green roofs which already exist. 
[revised manuscript text omitted]
 entire model domain, the sum of maximum water depths in the model domain is calculated for each simulation, which is called flood volume in the following. It has to be emphasized that these values do not represent the actual flood volume at a given time but rather an aggregated value that allows comparing the domain-wide effects of different settings and measures on the maximum water depths through one single value. Neumann et al. (2024) compared different scenarios with blue-green infrastructure in a similar way, while they included only cells with maximum depths higher than 10 cm. Table 1 lists the values

365   of flood volumes and relative differences for all simulations, including a few more simulations than the ones presented in the previous sections. First, the results of the simulations presented in the previous sections will be discussed, followed by the results of the additional simulations.

**Effects from rainfall sum and temporal distribution, infiltration, drainage, and retention roofs**

370   A first conclusion is that in the case of Euler-2 rainfall distributions, increases in the rainfall sum of 46% (Future 100a) and +123% (Strongest event), respectively, lead to increases in the flood volume of 61% (Future 100a) and 172% (Strongest event) compared to the Historical 100a event. The results show a non-proportional relation between rainfall sum and flood volume, with stronger increasing flood volume compared to the increase in the rainfall sum.

Comparing the flood volumes between Euler-2 and constant rainfall, constant rainfall leads to 48% (Historical 100a), 46%

375   (Future 100a), and 39% (Strongest event) smaller flood volumes than the respective event with a temporal rainfall distribution according to the Euler-2 design rainfall. These significant differences emphasize the importance of the temporal rainfall distribution and the need for sub-hourly rainfall intensities, while there is a tendency to decreasing importance with increasing rainfall sum. The simulations without infiltration show an increase in flood volume of 75% (Historical 100a) and 40% (Strongest event), respectively. Thus, even in such densely urbanized area (51% is considered to be completely sealed by

[Figure]

[Figure]

**Figure 11.** Temporal development of water depth and surface runoff within the Gleimtunnel without and with infiltration (top), the drainage system (middle), full retention roofs (top) - for Euler-2 rainfalls, maximum values along a longitudinal section through the Gleimtunnel; solid lines represent the reference case (with drainage and infiltration, without retention roofs)

[Figure]

[Figure]

380   buildings and roads, while the average sealing degree of unbuilt surfaces is around 26%), infiltration is strongly influencing
      the surface flooding, showing a decreasing importance with increasing rainfall sum. The combined effects of temporal rainfall
      distribution and infiltration, as well as time-dependent infiltration approaches could further be elaborated, but are beyond the
      scope of this study. The effect of the drainage system on the overall flood volume is reflected by reductions of 44% (Historical
      100a), 39% (Future 100a), and 35% (Strongest event), respectively, showing a slightly decreasing effectiveness with increasing

385   rainfall sum. As presented in the previous section, the drainage system is particularly successful in reducing the highest water
      depths at local hotspots. The retention roofs lead to reductions in flood volume of 37% (Historical and Future 100a) and 35%
      (Strongest event), respectively. As in the selected scenarios the retention roofs decreased the same percentage of the overall
      rain volume for all events, their effect on the flood volume for the different events is almost the same. For the Historical 100a
      event, the influence on the flood volume can be ranked as follows: 1) Infiltration (75%), 2) Temporal rainfall distribution (48%),

390   3) Drainage (44%), and 4) Retention roofs (37%). For the Strongest event, the ranking is: 1) Infiltration (40%), 2) Temporal
      rainfall distribution (39%), and 3) Drainage and Retention roofs (both 35%).

**Results from additional simulations**

      For the Historical 100a event, one simulation with a horizontal resolution of 2 m instead of 5 m was carried out. The overall

[revised manuscript text omitted]

---

## Author Response (AR1)

**Reply to Reviewer 1**

We would like to thank the anonymous reviewer for her/his positive evaluation and helpful comments. In the following we list the remarks of the review and add our answers below.

Review of Manuscript: "Extreme Precipitation and Flooding in Berlin under Climate Change and Effects of Selected Grey and Blue-Green Measures"

This manuscript addresses a timely and important topic, urban flooding under climate change, and evaluates the effectiveness of grey and blue-green adaptation measures for surface water flood risk management in Berlin. The study is well-structured and methodologically sound, incorporating climate projections, hydrodynamic modelling, and scenario-based analysis. The findings provide valuable insights for urban flood risk management and adaptation strategies, making the study relevant to a wide range of audience.

While the manuscript is well organised and written, and presents meaningful contributions, there are still areas that require further improvements before publication, outlined as follows:

**1. Uncertainty Quantification:**

The manuscript should provide a more detailed discussion of uncertainties associated with climate projections, model assumptions, and parameter sensitivity (e.g. sensitivity of flood simulation results to infiltration parameters/settings). Addressing these uncertainties will enhance the reliability of the findings.

Overall, more detailed discussions of uncertainties of climate projections, model assumptions, and parameter sensitivity have been included at corresponding subsections.

A sensitivity study for infiltration parameters/settings has been added by comparing simulations of varied datasets of saturated hydraulic conductivity and degree of surface sealing

We also adapted conclusions and included (beyond other paragraphs) the following:

"In this study the effect of climate change on extreme precipitation probability is studied and the resulting urban flooding is quantified. It has been taken care to include all relevant processes in the hydrodynamical simulations and represent them based on given data and knowledge. This includes a fully dynamic 2D robust shallow water model coupled to a 1D subsurface drainage model provided by the city's water company, and spatially distributed friction and infiltration parameter depending on the land use, soils, and degree of sealing." (...)

**and**

"The study is associated with a number of uncertainties, the most important being:

- The choice of the RCP8.5 climate scenario for the rainfall analysis and the fact that only one global-regional model combination was available for the study. This limitation affects the estimation of the strength of the precipitation change, that is at the upper range of changes reported by Hundhausen et al. (2024) for the area average in southern Germany.
- The highest uncertainties for flood calculations can be assumed to be associated with infiltration, which depends both on soil and surface sealing data. A sensitivity study reveals an uncertainty associated with infiltration of approximately 10% for the considered flood characteristics." (...)

Sensitivity analysis or confidence intervals could be included to better quantify the robustness of results.

An explanation on how the sampling uncertainties in the extreme precipitation a analysis were derived will be added: "The sampling uncertainty is determined using a bootstrap approach that randomly draws years with replacement.".

A sensitivity study on infiltration in terms of using 2 different datasets for the saturated hydraulic conductivity and different degrees of surface sealing has been added.

**2. Generalizability of Findings:**

Since the study focuses on Berlin, the authors should discuss how the findings could be applied to other cities with similar or different urban hydrology characteristics.

The findings generally apply also to other cities with similar hydrological characteristics. The increase in extreme rainfall intensities has been already observed in many regions of the world, and according to climate projections significant increases in the maximum daily rainfall intensity are expected for almost all land surfaces, while the percentages of increase depend on the emission scenario (IPCC, 2021). The effects of infiltration and the drainage system will also be similar depending on the design of the subsurface drainage system, given soils, and surface sealing conditions. The approach can be applied to any other city, while often the availability of high-quality and high-resolution data will be a limiting factor. Particularly, data or a model of the subsurface drainage system might often not be available. Reasonable drainage models could be generated by using AI and available data, e.g. on topography, roads, and sealed surfaces (Döring & Neuweiler, 2019). High-resolution DEMs based on LiDAR data as well as more detailed land use maps are getting more often available. Data on hydraulic soil properties are often a limiting factor, but associated uncertainties can be addressed through sensitivity analyses".

A brief comparison with existing studies from other regions would strengthen the broader applicability of the results.

The following comparison with other studies was added to the discussion section:

"Different studies showed similar effects of climate change and blue-green measures on urban flooding while using (partly) different methods. For example, Yang et al. (2024) demonstrated that climate change substantially increases the severity of urban flooding, particularly during short-duration extreme events. In contrast to our event-based, high-resolution 2D-1D hydrodynamic simulations, which used climate projections to derive extreme rainfall statistics and investigate spatial-temporal impacts, they coupled downscaled GCM scenarios with SWMM for long-term simulations of trends in annual runoff, flood volume, and the frequency of daily heavy rainfall in Dresden (Germany). Similarly, Li et al. (2024) combined CMIP6 climate projections with SWMM for a 6 km2 catchment in Haining City (China) and found that under SSP5-8.5, urban flood volumes from overflowing manholes could increase by more than 37% by the end of the century. Both studies, however, did not include 2D surface flow simulations, which enable high-resolution spatial analysis of flood depths and hotspots. Regarding blue-green measures, Liu et al. (2024) developed a dynamic framework coupling SWMM with GIS spatial analysis for Zhengzhou (China). They found that LID measures reduce runoff and inundation, but their effectiveness decreases with increasing rainfall intensity and is strongest under uniform rainfall patterns. This is consistent with our findings of reduced effectiveness under more extreme rainfall and a strong influence of temporal rainfall distribution. Similarly, Hua et al. (2020) showed for Chaohu City (China) that LID measures effectively reduce inundation, with higher performance under smaller storm magnitudes. Their model combined rainfall-runoff and 1D pipe flow with 2D overland flow, but the latter was only applied for surcharged runoff. By contrast, our study directly applies rainfall to the 2D grid and resolves surface—drainage interactions bi-directionally, capturing flow both before entering the drainage system and during overflow event."

**3. Data and Model Transparency:**

More details on the datasets, model calibration, and validation processes should be provided to ensure reproducibility.

Table with all used datasets has been added to section 2.3 for a better overview. Furthmore, the following paragraph has been added to section 3.2:

"The model is strongly based on physical relationships and available data for estimating model parameters. Model validation of the surface flow model was carried out by comparing results for other parts of the city with official flood maps from the municipality (Geoportal Berlin/Starkregen) as well as with simulations from other projects on urban flooding in Berlin (AMAREX; InnoMAUS). For the coupled model, plausibility checks have been carried out for selected heavy rain events comparing simulated time series of the water depth in a retention tank with observed data. The largest uncertainties might be associated to infiltration processes. In section 4.3.4, the constant infiltration capacity and degree of surface sealing are varied to assess their effect on the results."

How green roofs are represented/simulated in the model? Retention roofs are simulated by setting the rainfall in the input raster for all roof surfaces to zero. A full retention of all rainwater falling on roof surfaces is assumed. This has now been clarified in the methodology in section 3.2.

**Minor issues:**

Figures and Tables: Some figures and tables could be improved in clarity – consistent font size, format and clear legends should be used for all figures/tables.

The authors made efforts to improve the figures and tables with regard to consistent font size, format, and legends.

The manuscript should be carefully proofread for minor grammatical and typo errors. The authors have carefully proofread the final revised manuscript.

"The computation time increased by a factor of 3.3 from 2.4 hours (5 m resolution) to 7.8 hours (2 m resolution) while using 48 CPU cores (Intel Cascade Lake Platinum 9242) for both simulations." Did the hydrodynamic model simulations adopt adaptive timesteps? If yes, should be increasing factor of computational time from 2m to 5m resolution simulations ~10? When then resolution is doubled, the number of cells for calculation should be 4 times more and the time steps would be halved, leading to ~8 times of increase in runtime. So, increasing the resolution from 5m to 2m, the runtime should increase ~10 times?

Yes, in theory and with a linear solver behaviour, the runtime should increase by about 10 times. For larger simulations, we would also generally expect a stronger increase in the runtime, but not necessarily for smaller ones: If the system is not yet fully utilized anyway, then it scales even more favourably for the time being. In addition, the implementation of the solver as well as possible effects of the higher resolution on local flow characteristics affects the runtime. It is possible that the higher resolution has reduced/eliminated a numerical speed hotspot and therefore the time step has not decreased as much. Also large parts of dry areas during long periods of the simulation, which are not contributing to additional calculations, can contribute to a smaller increase in runtime. In our solver, the query for the vectorization (acceleration) of the calculation is grouped over a certain number of cells, i.e. it is not performed for individual cells. The higher the resolution, the more cells fit between two wet areas, so this may well lead to more skipped calculations.

The subsentence "showing a non-linear scaling behaviour" has been added to the paragraph.

Recommendation: Based on the above comments, I recommend moderate revisions before acceptance. Addressing the issues outlined above will significantly enhance the manuscript's clarity and quality. The study has strong potential for publication, provided the authors refine their discussion of uncertainties and generalisability.

**References**

AMAREX project, RPTU Kaiserslautern-Landau, URL: <a href="https://www.amarex-projekt.de/de">https://www.amarex-projekt.de/de</a> (last access 4 July 2025)

Döring, A., Neuweiler, I. (2019). Generation of Stormwater Drainage Networks Using Spatial Data. In: Mannina, G. (eds) New Trends in Urban Drainage Modelling. UDM 2018. Green Energy and Technology. Springer, Cham. <a href="https://doi.org/10.1007/978-3-319-99867-1">https://doi.org/10.1007/978-3-319-99867-1</a> 99

Fowler, H.J., Lenderink, G., Prein, A.F. et al. (2021). Anthropogenic intensification of short-duration rainfall extremes. Nat Rev Earth Environ 2, 107–122. <a href="https://doi.org/10.1038/s43017-020-00128-6">https://doi.org/10.1038/s43017-020-00128-6</a>

Geoportal (2025). Hinweiskarte Starkregengefahren, URL: <a href="https://www.geoportal.de/map.html?map=tk\_04-hinweiskarte-starkregengefahren-be-bb">https://www.geoportal.de/map.html?map=tk\_04-hinweiskarte-starkregengefahren-be-bb</a> (last access 04 July 2025)

Hua, P., Yang, W., Qi, X., Jiang, S., Xie, J., Gu, X., Li, H., Zhang, J., and Krebs, P. (2020): Evaluating the effect of urban flooding reduction strategies in response to design rainfall and low impact development, Journal of Cleaner Production, 242, 118 515, <a href="https://doi.org/https://doi.org/10.1016/j.jclepro.2019.118515">https://doi.org/https://doi.org/10.1016/j.jclepro.2019.118515</a>

Hundhausen, M., Feldmann, H., Kohlhepp, R., and Pinto, J. G. (2024). Climate change signals of extreme precipitation return levels for Germany in a transient convection-permitting simulation ensemble, International Journal of Climatology, 44, 1454 – 1471, <a href="https://api.semanticscholar.org/CorpusID:270475096">https://api.semanticscholar.org/CorpusID:270475096</a>

Inno\_MAUS project, University of Potsdam, URL: <a href="https://www.uni-potsdam.de/de/inno-maus/">https://www.uni-potsdam.de/de/inno-maus/</a> (last access 4 July 2025)

IPCC (2021). Climate Change 2021: The Physical Science Basis. Contribution of Working Group I to the Sixth Assessment Report of the Intergovernmental Panel on Climate Change[Masson-Delmotte, V., P. Zhai, A. Pirani, S.L. Connors, C. Péan, S. Berger, N. Caud, Y. Chen, L. Goldfarb, M.I. Gomis, M. Huang, K. Leitzell, E. Lonnoy, J.B.R. Matthews, T.K. Maycock, T. Waterfield, O. Yelekçi, R. Yu, and B. Zhou (eds.)]. Cambridge University Press, Cambridge, United Kingdom and New York, NY, USA, In press. <a href="https://doi.org/10.1017/9781009157896">https://doi.org/10.1017/9781009157896</a>

Li, Y., Wang, P., Lou, Y., Chen, C., Shen, C., and Hu, T. (2024): Assessing urban drainage pressure and impacts of future climate change based on shared socioeconomic pathways, Journal of Hydrology: Regional Studies, 53, 101 760,665 https://doi.org/https://doi.org/10.1016/j.ejrh.2024.101760.

Liu, C., Xie, T., Yu, Q., Niu, C., Sun, Y., Xu, Y., Luo, Q., and Hu, C. (2024): Study on the response analysis of LID hydrological process to rainfall pattern based on framework for dynamic simulation of urban floods, Journal of Environmental Management, 351, 119 953, <a href="https://doi.org/https://doi.org/10.1016/j.jenvman.2023.119953">https://doi.org/https://doi.org/10.1016/j.jenvman.2023.119953</a>

Yang, W., Zhao, Z., Pan, L., Li, R., Wu, S., Hua, P., Wang, H., Schmalz, B., Krebs, P., and Zhang, J.(2024): Integrated risk analysis for urban flooding under changing climates, Results in Engineering, 24, 103 243, <a href="https://doi.org/https://doi.org/10.1016/j.rineng.2024.103243">https://doi.org/https://doi.org/10.1016/j.rineng.2024.103243</a>

**Reply to Reviewer 2**

We would like to thank the anonymous reviewer for her/his positive evaluation and helpful comments. In the following we list the remarks of the review and add our answers in bold.

The manuscript presents a study on urban flooding and the influence of climate change on heavy precipitation and the consequent urban flooding. Moreover, it evaluates factors and measures to reduce flooding, like the effect of infiltration on existing unsealed areas on flooding, the actual effect of the sewer system, which is often neglected in the derivation of urban flood hazard maps, and the effect of storage roofs.

The manuscript is well written and structured, and provides useful insights into potential adaptation measures. However, there are a few issues to be dealt with to improve the manuscript even further:

The scenario analyzing the effect of infiltration is only "retrospective/negative", means that only the effect of infiltration on flooding from existing unsealed areas is considered. While this is a useful insight into the dimension of flood reduction by the existing unsealed areas, it does not show the potential of further de-sealing for flood reduction. This would be an important information to the city authorities within the frame of the flood management and climate change adaptation, because de-sealing is an essential element of the sponge city concept. Thus, I suggest to include also a scenario with additional infiltration by de-sealing where it is likely possible in reality (e.g. parking lots, pedestrian areas etc.). This would be much more helpful for the city and the wider audience, because it shows the potential of what can be achieved with additional desealing, in addition to showing that the current infiltration is effective in reducing urban flooding. Particularly because the other option of increased drainage, i.e. increasing the sub-surface drainage capacity (sewer system), is not that easily achievable in practice, and very costly.

We decided to compare a simulation with infiltration settings representing the actual state to the one of complete sealing, which already provides useful information on the effect of infiltration. Constructing a realistic de-sealing scenario is beyond the scope of this study, but we added now more simulations with different infiltration settings investigating the sensitivity to the hydraulic conductivity and sealing degree. Here, also one scenario of decreased de-sealing of the street areas (which include parking lots and sidewalks) as suggested by the reviewer has been included. For future work, effects of realistic de-sealing scenarios would be very interesting, especially when conducted by the local authorities that have better data on potential areas for de-sealing measures.

The conclusions are very short. Please elaborate more on recommendations for flood proofing the city based on your results, and also on the transferability to other cities. This can be done on general reasoning and literature.

The conclusions have been revised. The transferability to other cities has been discussed the discussion in "The findings generally apply also to other cities with similar hydrological characteristics. The increase in extreme rainfall intensities has been already observed in many regions of the world, and according to climate projections significant increases in the maximum daily rainfall intensity are expected for almost all land surfaces, while the percentages of increase depend on the emission scenario (IPCC, 2021). The effects of infiltration and the drainage system are expected to also be similar depending on the design of the subsurface drainage system, soils, surface sealing conditions, and topography. The approach can be applied to any other city, while often the availability of high-quality and high-resolution data will be a limiting factor. Particularly, data or a model of the subsurface drainage system might often not be available or accessible. Reasonable drainage models could be generated by using AI and available data, e.g. on topography, roads, manholes, and sealed surfaces (Döring and Neuweiler, 2019). Highresolution DEMs based on LiDAR data as well as more detailed land use maps are getting more often available. Data on hydraulic soil properties are often a limiting factor, but associated uncertainties can be addressed through sensitivity analyses."

I see the way the flood volume is calculated problematic. What you actually calculate is the sum of maximum water depths, which is definitively not an estimation of the actual flood volume. Sure, you can show how this reduced by the individual measures, but this is not showing the reduction in flood volume and thus the percentages you provide have limited meaning. I recommend to use the sum of the modelled surface water at the end of the rain event. This is a much more realistic estimation of the actual surface flood volume, that considers the drainage by infiltration and the sewer system.

We thank you for raising this point. We used the sum of maximum water depths to compare the effects of different settings on the worst-case situation (max. water depth) in the entire domain by one single value. But to call this "flood volume" is indeed misleading. We included now a comparison of flood volumes after 30 minutes (time of the peak water depths in the Gleimtunnel), and 120 minutes (end of the simulation time). The results after 60 minutes (end of rainfall) as suggested by the reviewer have also been investigated, but where not included here, since the results are between the two other ones and the selected timesteps represent a longer period and important moments during the flood development.

I cannot really follow how the sub-hourly rainfall intensities are derived (section 4.3 and Fig. 5). Please revise this section to make this clearer.

We suggest to rephrase the paragraph (lines 221-236) to:

"Since the 100-year events for the near and far future simulations were very similar and the near future event even a bit stronger, only the 100-year event of the near future period was chosen for the hydrodynamic simulations. For the construction of the Euler-2 design rainfall, intensities for duration classes of less than one hour are needed. As already explained in section 4.1, the precipitation output from the climate simulations is one hour and extrapolating sub-hourly intensities from the modeled probability distribution leads to strong overestimations of sub-hourly intensities compared to values from KOSTRA. To overcome this problem, a Euler-2 time series is constructed by scaling the statistical hourly and sub-hourly 100-year return values from KOSTRA-DWD-2020. The scaling factor is applied to each time step and calculated as the ratio between the 1-hour rainfall sum of the respective event (as listed above) and the 1-hour rainfall sum of the 100-year event from KOSTRA-DWD-2020 (48.5 mm). The resulting scaling factors are 0.98 (47.7/48.5) for the Historical 100a event, 1.44 (69.8/48.5) for the Future 100a event, and 2.20 (106.7/48.5) for the Strongest event. For the Strongest event, the maximum intensity calculated with this approach is 45 mm in 5 min (9 mm/min). This is considered to be unrealistically high as the maximum 5-min rainfall measured at the station in Berlin-Dahlem during the period 1979 to 2023 was 12.3 mm. For an additional comparison with less extreme maximum intensities, all selected events were also simulated with a constant rainfall intensity. This also allows to analyze the influence of the temporal rainfall distribution. Fig. 5 shows the Euler-2 and constant rain events for the three aforementioned rainfall sums."

There are more comments/questions in the annotated pdf.

As there was only one combination of global-regional climate model available, it would be very useful to provide an assessment of how representative the result on the increase of rainfall intensities in future is/can be expected in relation to a full ensemble of climate projection by different models. Particularly the finding that the far-future projection gives reduced rainfall intensities compared to the near future is unexpected, as you also state. This can be done by literature references. If it can be shown that this result is in line with other studies/projections, this would substantiate your findings and conclusions considerably.

Please find our answers and suggestions for a revision below in points L 205 and L 488.

Next to this I made some additional minor comments or raised questions in the annotated pdf of the manuscript.

**Further comments from the annotated pdf:**

L 50: Some details on this would be helpful here. A figure showing the Euler-2 sub-hourly rainfall distribution would also be helpful. We suggest to alter the description as follows: "The Euler-2 evolution is characterized by gradually increasing intensities,

reaching the 5-minute maximum after 20 minutes, and a subsequent strong decrease down to a small residual amount (as illustrated in section 4.3, Fig.5). The time series is constructed using hourly and sub-hourly, statistical precipitation intensities from the coordinated heavy precipitation regionalization and evaluation (KOSTRA, 2020) of the German Weather Service (DWD; see section 2)"

L. 154: surface or sewer drainage? Sewer drainage catchment; has been adapted.

L. 166: I am missing a description of how the buildings are treated in both models. Are they excluded from hms++ model, or considered as building blocks in the DEM? They are considered with increased elevation in the DEM. This is now clearly described in the model setup.

What happens to rainfall falling on roofs? Is this added to the sub-surface drainage and simulated as such in SWMM?

No, it is added directly to the surface as the buildings are represented just by increasing the elevation. This has been only mentioned in the discussion, now it is also explained in the model setup.

Information on this is essential for interpreting the no-subsurface-drainage and the retention roof scenarios.

L.173: for the study area or whole Berlin? For the study area; clarification was added.

L. 191: where is this shown? The reference has been added to Fig. 3a

Fig. 3 Please include also the KOSTRA area-averaged estimations of the 100-year event for other durations. This would give a more encompssing view of the realiability of the historic simulations vs. the observation-based KOSTRA data; it should also show then the described deviation of the d-GEV fitting from KOSTRA for the durations < 1 h

We will add further KOSTRA values and change the y axis label for (b) as suggested to "Intensity Change".

L. 198: +/- 1 Kelvin for a climatological mean temperature is quite a lot. Can you comment on this? Has these implications on the results and conclusions?

For the hydrodynamic simulations conducted in our study temperature does not play a role. As the extreme precipitation statistics agree well with observations we are confident that shortcomings of other variables will have no significant implications for the results and conclusions. For other impact studies relying on realistic temperatures a bias adjustment would be needed.

L. 205: can this be physically explained? or is it just a random result? would different slightly different realizations of CCLM-CPS or MIROC5 (nudging with different starting points or varying parameter sets, etc.) give the same or different results?

Answering this question could get lengthy, I know, but the presented result is counter-intuitive, on the first glance at least. Thus a few sentences to help interpreting/judging the findings would be very helpful. Citing other studies with the same or similar results might help here.

**Or is this an artefact of the d-GEV fitting?**

This is a random result and caused by the strong variability in the simulations (variability of extremes is predicted to increase under climate change conditions). The result demonstrates that 30 years are a rather short period for a robust GEV analysis aimed at estimating 100-year return periods. The variability can be seen e.g. in Fig. 7 of Hundhausen et al. (2024). Hundhausen et al. (2024) analysed precipitation extremes in an ensemble of continuous simulations at convection permitting resolution covering southern Germany. We will extend the description to "This is a result of the high variability in extreme precipitation that is superimposed on the overall increase (e.g. Hundhausen et al. 2024) and illustrates the need for long simulations for robust estimates."

L 226: Can this be compared/used? Does KOSTRA provide sub-hourly rainfall distributions? To my knowledge, KOSTRA presents the statistical evaluation of rainfall frequencies of different duration, which is not necessarily a quantitative estimation of the sub-hourly rainfall distribution of an 1-hour event. Please provide explanation. This means you scale the 5-min rainfall estimates to match the 1-hour 100-year event sum of KOSTRA, correct? Overall, this whole section is not well understandable and needs revision. A graphical sketch of the workflow would certainly help.

The Euler-2 temporal evolution is constructed using the statistical intensities (not a realistic sub-hourly disaggregation of the hourly event). KOSTRA does provide the necessary statistical sub-hourly statistical intensities.

It is correct that we scale the 5-min rainfall estimates from KOSTRA. As a scaling factor we use the ration between the simulations and KOSTRA determined for the 100-year hourly event. As we already have a high number of figures, we would like to avoid adding another one. Please see our suggestion to rephrase the paragraph lines 221-236 above.

L. 293: What would be realistic for retention roofs? can they be realized on the typical gable roofs in Berlin? My assumption is not, but only flat roofs can be converted to retention roofs. Is this correct?

A few words on the plausibility of the assumption that all roofs can be converted to retention roofs are advisable.

It is correct, that only flat roofs or roofs with a small slope can be converted to retention roofs.

The following paragraph has been adapted and shifted in front of showing the results:

"These simulations are not intended to represent realistic scenarios, but rather to explore the range of possible system behaviors and assess the sensitivity of flooding dynamics to infiltration, drainage, and retention roofs. For example, the case of full retention of the rain on all roof surfaces can be seen as a best-case scenario. While the complete retention during all considered heavy rain events could hardly be realized by modifying all roofs in the study area to become green or retention roofs, it could - in theory - be achieved with combinations of green and/or retention roofs and cisterns for all buildings in that area. The stored water could also help conserve drinking water by being used for non-potable purposes, such as toilet flushing or irrigation. A realistic scenario of considering only suitable roofs for retention roofs, for example based on their inclination, is beyond the scope of this study."

So, instead of retention/green roofs, the retention of the rainwater could also be achieved by draining the water from the roofs into (subsurface) cisterns. An analysis of the potential for retention roofs in the study area is beyond the scope of this study, as we focused to investigate the theoretically maximum possible effect by capturing rain water falling on roof surfaces.

L.361: That's absolutely correct and opens the question why not the actual flood volume is used. This can be easily calculated summing up the surface water depths at the end of the rainfall (i.e. t = 60 min). This would be a realistic number considering the direct drainage of rainfall. I recommend to use this value in the analysis.

As mentioned in an answer to an earlier comment, we adapted the calculation of the flood volume and compare the volumes at different time steps. We selected now the timesteps of 30 and 120 minutes to show the flood volumes during the peak (max rainfall and max water depth at the Gleimtunnel) and at the end of the simulation time. We also checked the volumes at the end of the rainfall event (60 min), which are between the two others. So cover different situations during the flood event, we decided to show the results after 30 and 60 minutes.

L.371: These numbers are based on that unrealistic calculation of flood volume, thus they have very limited practical meaning, unless you show that the flood volume as you calculated is the same as the flood volume calculation I suggest above. If you use the actual flood volume as I suggest, you can avoid a discussion like this and the volume evaluation in relation to rainfall sum would be much more robust and meaningful.

This comment refers to all other numbers in this volume comparison section.

As mentioned in an answer to an earlier comment, we will adapt that and compare the volumes at different time steps.

L433: But you used data from the soil survey, thus the simulation of the infiltration based on the surveyed conductivities should not be that uncertain.

But you are right, for the simulation of de-sealing measures this is an issue, as the soil under buildings is not surveyed. But this has not been done in this study.

You may want to change the statement towards this argument.

The available soil data is relatively coarse (on a block level), so local details might not be captured. kf values as well as (un-)sealing degrees come along with uncertainties. This is addressed now in a short study in Section 4.3.4. varying kf values (as constant infiltration capacity) and sealing degrees, together with one scenario with increased unsealing as proposed by the reviewer.

L.435: I made a comment about this in the method section. It is important to state this earlier in the methods, not only here in the discussion.

We will now state this already in the section about the model setup:

"In the model, no roof drainage directly into the sewer system is represented. Since buildings are characterized by their elevated position, rain falling on the roofs flows off the buildings, either into backyards or onto street surfaces, from where it eventually enters the sewer system via the street inlets."

L. 436: is this simulated like this in the hms++ model? this also needs to be stated in the method section.

We clarifed this in the section about the model setup.

Moreover, the statement that "most" of intense rainfall is overspilling the roof drainage is debatable. Is there any evidence for this, e.g. design values for roof drainage vs. rainfall? To my knowledge roof drainage is as designed to the same standard as street drainage.

Means, neglecting roof drainage is actually the same as assuming no subsurface drainage for the whole area, as it is usually done in pluvial flood mapping. But drainage is important, as you correctly state in the introduction. Thus, why is roof drainage neglected? Wouldn't it be closer to reality to consider 100% rood drainage than none?

The available drainage model data does not include the roof drainages. We could make assumptions on it, but this would mean a lot of effort to connect all roof drainages correctly to the subsurface drainage network in the model. In our case, the rain is flowing from the roofs to the adjacent areas, in case of roads, the water can be drained into the subsurface drainage system by street inlets, which can be considered as good approximation of the actual roof drainage directly into the subsurface drainage system. Some of the runoff also flows to backyards, where no connection to the subsurface drainage system is present in the model. This part of surface flooding might indeed be to some extent an overestimation, as at least some part of the water would be drained into the subsurface drainage system instead of reaching the backyard. At the same time, this means some kind of underestimation of water present in the subsurface drainage

system and therefore also combined sewer overflow. We will discuss these effects due to the simplified assumptions without roof drainage. A inclusion of roof drainage into the subsurface drainage system is not feasible within this study.

We added also the following paragraph to the discussion section:

"One simplification of the model is that direct drainage from roofs into the drainage system was not included. Instead, runoff from rainfall on the roofs flows to adjacent areas. For streets, water can enter the subsurface drainage system via street inlets, which provides a reasonable approximation of actual roof drainage directly into the system, noting that this is effective for only part of the rainfall volume. Some runoff also flows into backyards, where no connection to the subsurface drainage system is represented in the model. This may lead to a slight overestimation of surface flooding, as some of the water would likely be drained into the subsurface system rather than reaching the backyard. In contrast, this implies a potential underestimation of water in the subsurface drainage system and, consequently, in the combined sewer overflow."

L. 488: add this paragraph to the paragraph stating uncertainties above.

the conclusion is based on the assumption that the single used global-regional climate model combination is representative for the actual climate development/the usually used ensemble (median). You should prove that this increase in heavy precipitation is very likely in the text above (literature), that the model represents this well, and state this here to support your conclusion.

We suggest to extend the paragraph:

"The study is associated with a number of uncertainties, the most important being:

- The choice of the RCP8.5 climate scenario for the rainfall analysis and the fact that only one global-regional model combination was available for the study. This limitation affects the estimation of the strength of the precipitation change, that is at the upper range of changes reported by Hundhausen et al. (2024) for the area average in southern Germany.
- The highest uncertainties for flood calculations can be assumed to be associated with infiltration, which depends both on soil and surface sealing data. A sensitivity study on sealing degree and saturated hydraulic conductivity shows that uncertainties in infiltration translate to roughly 10% variation in the analyzed flood characteristics.

The fact that extreme precipitation intensifies under global warming conditions can be regarded as undisputed and has been recognized by a large number of studies (IPCC, 2021; Fowler et al., 2021). Irrespective of the various uncertainties associated with this analysis, the investigation shows very clearly that urban flood risk maps must take climate change into account in order to adequately contribute to the protection of the

population. Risk maps based on past observations with the assumption of a stationary climate will be soon outdated.

**References**

Fowler, H.J., Lenderink, G., Prein, A.F. et al. (2021). Anthropogenic intensification of short-duration rainfall extremes. Nat Rev Earth Environ 2, 107–122. https://doi.org/10.1038/s43017-020-00128-6

IPCC (2021). Climate Change 2021: The Physical Science Basis. Contribution of Working Group I to the Sixth Assessment Report of the Intergovernmental Panel on Climate Change[Masson-Delmotte, V., P. Zhai, A. Pirani, S.L. Connors, C. Péan, S. Berger, N. Caud, Y. Chen, L. Goldfarb, M.I. Gomis, M. Huang, K. Leitzell, E. Lonnoy, J.B.R. Matthews, T.K. Maycock, T. Waterfield, O. Yelekçi, R. Yu, and B. Zhou (eds.)]. Cambridge University Press, Cambridge, United Kingdom and New York, NY, USA, In press. https://doi.org/10.1017/9781009157896

Hundhausen, M., Feldmann, H., Kohlhepp, R., and Pinto, J. G. (2024). Climate change signals of extreme precipitation return levels for Germany in a transient convection-permitting simulation ensemble, International Journal of Climatology, 44, 1454 – 1471, <a href="https://api.semanticscholar.org/CorpusID:270475096">https://api.semanticscholar.org/CorpusID:270475096</a>